# Toward Xeno-Free Differentiation of Human Induced Pluripotent Stem Cell-Derived Small Intestinal Epithelial Cells

**DOI:** 10.3390/ijms23031312

**Published:** 2022-01-24

**Authors:** Jaakko Saari, Fatima Siddique, Sanna Korpela, Elina Mäntylä, Teemu O. Ihalainen, Katri Kaukinen, Katriina Aalto-Setälä, Katri Lindfors, Kati Juuti-Uusitalo

**Affiliations:** 1Faculty of Medicine and Health Technology, Tampere University, 33014 Tampere, Finland; jaakko.saari@tuni.fi (J.S.); fatima.siddique@tuni.fi (F.S.); sanna.korpela@tuni.fi (S.K.); elina.mantyla@tuni.fi (E.M.); teemu.ihalainen@tuni.fi (T.O.I.); katri.kaukinen@tuni.fi (K.K.); katriina.aalto-setala@tuni.fi (K.A.-S.); katri.lindfors@tuni.fi (K.L.); 2Department of Internal Medicine, Tampere University Hospital, 33521 Tampere, Finland; 3Heart Hospital, Tampere University Hospital, 33521 Tampere, Finland

**Keywords:** small intestinal epithelial cell, induced pluripotent stem cell, iPSC, serum-free culture, KO-SR, recombinant laminin, iPSC-SIEC, 2D iPSC-SIEC

## Abstract

The small intestinal epithelium has an important role in nutrition, but also in drug absorption and metabolism. There are a few two-dimensional (2D) patient-derived induced pluripotent stem cell (iPSC)-based intestinal models enabling easy evaluation of transcellular transport. It is known that animal-derived components induce variation in the experimental outcomes. Therefore, we aimed to refine the differentiation protocol by using animal-free components. More specifically, we compared maturation of 2D-cultured iPCSs toward small intestinal epithelial cells when cultured either with or without serum, and either on Geltrex or on animal-free, recombinant laminin-based substrata. Differentiation status was characterized by qPCR, immunofluorescence imaging, and functionality assays. Our data suggest that differentiation toward definitive endoderm is more efficient without serum. Both collagen- and recombinant laminin-based coating supported differentiation of definitive endoderm, posterior definitive endoderm, and small intestinal epithelial cells from iPS-cells equally well. Small intestinal epithelial cells differentiated on recombinant laminin exhibited slightly more enterocyte specific cellular functionality than cells differentiated on Geltrex. Our data suggest that functional small intestinal epithelial cells can be generated from iPSCs in serum-free method on xeno-free substrata. This method is easily converted to an entirely xeno-free method.

## 1. Introduction

Proper functionality of small intestinal epithelium is important in nutrition, but also in drug absorption and metabolism. Cell-based intestinal models, an alternative to animal models, have been widely utilized to investigate human small intestine functionality and in drug screening studies [1,2]. From among the currently used models, Caco-2, a colonic adenocarcinoma-derived immortal cell line, has been the gold standard for intestinal research for decades [3,4]. When reaching confluency, Caco-2 culture polarizes spontaneously to form a monolayer, which is easy to manipulate and to access both from apical and basal sides. Caco-2 has been valuable in high throughput screening of drug permeability and the identification of various crucial transporters, inhibitors, and inducers of the small intestinal epithelium [5]. However, the Caco-2 cell line displays major limitations in the expression of some transporters such as peptide transporter 1 (PEPT1), and drug metabolizing enzymes such as cytochrome P450 CYP3A4, in comparison to intestinal mucosa in vivo [3].

Primary cells offer the most desirable and presentative material for studying the intestinal epithelium in vitro. Unfortunately, the yield is of these cells is usually rather low, they have a short lifespan in the culture and have problems with cell orientation and contamination by microbes or stromal cells [6,7].

Intestinal organoids, or “mini-guts”, are three-dimensional (3D) self-organizing clusters formed either from intestinal stem cells or pluripotent stem cells, and since their discovery by Sato et al. in 2009 [8,9], they have been extensively used in intestinal research [10,11,12,13]. The organoids mimic the crypt-villus architecture, functionality, and overall heterogeneity of the intestinal epithelium remarkably well [10,11,12,13]. Initially the organoids were only grown in 3D, which are closed structures having apical, food-ingesting sides inside the lumen. Currently, organoids can be replated on 2D surface to form easily accessible and functional small intestinal epithelial culture [14,15,16]. When 3D-cultured organoids are cultured in 2D, the highly polarized and columnar intestinal epithelial cells form cuboidal, or even thinner epithelial cell layers [16].

In addition to the organoids, induced pluripotent stem cells (iPSCs) can be cultured in 2D and differentiated to generate all small intestinal epithelial cell types (SIECs) [17,18,19,20,21,22]. In 2D cultures, the cells are grown on flat porous surface, which allows for the handling of the culture both from apical and basal sides of the cells. As 2D culture is accessible from both sides of the cells, i.e., from both sides of the culture and thus is advantageous in high throughput pharmacokinetic testing [23,24]. In addition, the possible genetic disease predispositions are present in the resulting SIECSs and allow for studying of the possible effects of genetic factors in the function of the iPSC-differentiated cells in vitro. All published methods are based on the developmental pathway of embryo- and organogenesis in vitro starting from the establishment of definitive endoderm (DE), followed by posterior definitive endoderm (PDE) induction, and finally differentiating cells toward SIECs [17,18,19,20,21,22].

We aimed to refine 2D iPSC SIECs differentiation protocol which uses fewer animal-derived components than current protocols. More specifically, we compared maturation of 2D-cultured iPCSs toward SIECs on collagen- or on xeno-free recombinant laminin-based substrata. In addition, we optimized the differentiation toward definitive endoderm definitive endoderm when cultured either with fetal bovine serum (FBS) or knock-out rerum replacement (KO-SR), and with different induction times and concentrations of Wnt3a and CHIR.

## 2. Results

### 2.1. Effects of Serum Replacement and Wnt-Signaling Inducers on DE-Differentiation

First, we assessed whether FBS or KO-SR were any different in their capacity to induce differentiation of iPSCs toward DE. In this analysis, cells were grown for 72 h in either FBS or KO-SR with or without the presence of Wnt-inducing agents Wnt3 or CHIR, and the expression of differentiation markers SOX17 and FOXA2 were detected by using immunostaining. Here, in comparison to cells grown in the presence of 0.5% FBS, the cells in higher (2%) FBS concentration, the cells proliferated more rapidly, especially in higher (60 ng/mL and 90 ng/mL) Wnt concentrations (Figure 1a). In addition, in higher (2%) FBS concentrations, the lower number of SOX17 and FOXA2 was observed (Figure 1a).

When FBS was omitted and replaced with a 0.2% and 2% KO-SR (24 h–48 h 0.2% and 48 h–72 h 2%), the number of cells determined as the number of DAPI-stained nuclei, and cells the positive for both definitive endoderm markers, SOX17 and FOXA2, was elevated in all samples studied (Figure 1b). Similarly, adding Wnt-inducer rhWnt3 at concentrations of 30 and 60 ng/mL yielded comparable amounts of double-positive cells independent of the exposure time (Figure 1b). CHIR99021 at both tested concentrations produced similar numbers of SOX17 and FOXA2 double-positive cells as Wnt when incubated for 24 h (Figure 1b). If CHIR treatment was extended to 72 h, the overall number of cells decreased along with the number of SOX17 and FOXA2-positive cells (Figure 1b). Furthermore, the number of cells evincing micronuclear bodies, increased. If Activin A, a known TGFβ and nodal signaling inducer [25], was used alone without any Wnt-signaling inducer, there were no SOX17-positive cells (Figure 1b).

The gene expression studied with quantitative real time PCR (qRT-PCR) (Figure 1c) was in line with the immunofluorescence staining results (Figure 1a,b); the cells induced to differentiate in KO-SR conditions generally expressed *SOX17* more in all samples studied than cells differentiated with FBS (Figure 1c). The only exception was KO-SR + Act-A while expression of *SOX17* was noticeably lower. The lowest mesodermal marker *TBXT/ BRACHYURY* expression was achieved when cells were differentiated with 30 ng/mL WNT3a (3 days) and 2% FBS. In these conditions, the expression of *SOX17* was almost non-existent (Figure 1c). The second lowest expression was achieved when differentiation was conducted with KO-SR and 2.5 µM or 5 mM CHIR for one day or 30 ng/mL WNT3a for three days with 0.5% FBS. High expression of *TBXT* when CHIR was used with FBS indicates differentiation into mesoderm cells. Expression of pluripotency marker *NANOG* was generally higher in cultures where differentiation was induced with KO-SR, than with FBS, while the expression of *OCT4* was low in all conditions apart from KO-SR and Act-A. The extended treatment or treatment with high concentrations (5 µM) of CHIR lowered the cell viability; thus, CHIR was later used in lower concentrations (2.5 µM CHIR) for one day. The results above suggested the differentiation without serum gave higher number of SOX17-positive cells, the FBS was omitted, and replaced with KO-SR in following tests.

### 2.2. Effects of Substrata on the Differentiation toward Definitive Endoderm

Animal-derived cell culture products may vary from lot to lot. Therefore, we aimed as xeno-free culture substrata, and compared commonly used substrata, Geltrex (a basement membrane extracted from murine Engelbreth–Holm–Swarm (EHS) tumors that contain laminin, collagen IV, entactin, and heparin sulfate proteoglycans) with human recombinant laminin 111, with human fibronectin and with a combination of latter two. iPSCs were differentiated toward definitive endoderm on those four substrata for 72 h.

In immunofluorescence staining, Geltrex and laminin surface yielded higher numbers of cells, and greater numbers of SOX17 and FOXA2-positive cells than fibronectin or laminin and fibronectin (Figure 2a). Fibronectin and a combination of laminin and fibronectin substrata resulted in more pronounced SOX17 staining but fainter FOXA2 staining than Geltrex or laminin alone.

Next, the effects of growth substrata protein on the differentiation were analyzed in induction experiments with Wnt3a and CHIR (Figure 2b). The differentiation gene expression (*SOX17*, *TBXT/BRACHYURY, OCT3/4* and *Nanog*) levels determined by qPCR were normalized to expression levels acquired on Geltrex, (and thus not shown in Figure 2b). A small increase in the expression of *SOX17* was observed on all coatings compared to Geltrex when 2.5 µM CHIR was used as an inducer (Figure 2b). The expression was also elevated with 1d 30 ng/mL Wnt3a condition on laminin and fibronectin surfaces. The expression of *TBXT/BRACHYURY* was also increased on all coatings in CHIR conditions but only to a minor degree on laminin. In the Wnt3a condition, the expression of *TBXT/BRACHYURY* expression decreased in cells cultured on pure laminin 111 or fibronectin but increased in cells cultured on laminin 111-fibronectin mixture. However, the expression of *TBXT/BRACHYURY* was considerably higher in the original Geltrex tests when calibrated against iPSC sample (Figure 2b) than in CHIR differentiated samples. Thus, a direct comparison cannot be made between CHIR and Wnt-treatments. The expression of pluripotency markers *OCT3/4* and *Nanog* was modest across the different coatings and conditions tested. In this experiment, the cells differentiated with 2.5 µM CHIR on laminin surface had the lowest pluripotency marker gene (*OCT3/4* and *NANOG*) gene expression. Cells grown on Geltrex had more intense FOXA2 staining than cells grown on fibronectin alone or laminin 111-fibronectin mixture (Figure 2a). The percentage of SOX17 positive cells, was markedly lower in cells grown on fibronectin alone and on laminin 111-fibronectin mixture than on Geltrex (Figure 2a). As immunofluorescence data generated with the 2.5 µM CHIR and 3d Wnt3A, the immunofluorescence data (Figure 2a) are in line with the gene expression data (Figure 2b). As fibronectin and laminin 111-fibronectin coatings resulted in low SOX17 expression, further experiments were carried out with Geltrex and laminin substrata.

### 2.3. Induction toward an Instestinal Lineage

DE-differentiated iPSCs were induced to differentiate toward intestinal lineage (posterior DE) by using LY2090314 or recombinant human (rh) bFGF. LY2090314 is a known glycogen synthase kinase-3 inhibitor, which can activate Wnt/β-catenin signaling [19]. LY2090314 is also shown to promote *CDX2* expression, which is a marker of posterior definitive endoderm [19]. The induction was assessed by staining the cells against an intestinal lineage marker CDX2 (Figure 3a) and the exposure times were held constant in all samples inspected. LY2090314 resulted more intense staining of CDX2 than bFGF (Figure 3a). After LY2090314 treatment, there were no OCT3/4 and only few NANOG-positive cells, while after bFGF, there was quite a high number of NANOG-positive cells. Induction with LY2090314 resulted in the same number of CDX positive cells but clearly smaller number of SOX17-positive cells than bFGF when cells were grown on Geltrex (Figure 3b). There was no difference in the number of double-positive cells when cells were induced to differentiate with bFGF on Geltrex (Figure 3b). The same effect of increased SOX17 expression seen with bFGF on Geltrex in immunofluorescence staining, was seen in the gene expression analysis (Figure 3c). When the PDE-differentiated iPSCs were detached and further plated either on Geltrex or laminin 511, only bFGF supplemented cells adhered in sufficient quantities to the transwell insert to form confluent culture which was able to differentiate and mature (data not shown). As the number of post-passage viable cells was low after LY2090314 supplementation, all further experiments were performed using only bFGF.

### 2.4. Induction toward Small Intestinall Epithelial Cells

iPSC-derived posterior definitive cells were further differentiated into small intestinal epithelial cells SIECS with recombinant wnt3A-EGF-Noggin-R-Spondin (WENR) containing medium on either Geltrex or laminin 511-substrata. At first cells proliferated actively and formed a uniform monolayer (data not shown). After 28 days of culture, there was some heterogeneity in the cultures (Figure 4a); the cells had acquired columnar morphology on most areas, although locally cuboidal-shaped cells were detected. While protein substrata had only minor effects on the x/y morphology of the epithelium, the thickness of the subepithelial extracellular matrix varied more on Geltrex, resulting to some areas thicker extracellular matrix than on laminin (Figure 4a, arrowhead in Chromogranin on Geltrex culture, and arrow in the same but on laminin). On both, substrata enterocyte specific protein PEPT1, was mainly localized on the apical side of the cells although there was some staining within the cytoplasm (Figure 4a). Villin staining was found in almost all iPSC-SIEC cells cultured on both substrata. In iPSC-SIEC cultured on Geltrex the staining was found to be scattered within the cell, whereas in iPSC-SIEC cultured on laminin, the staining was more concentrated on the apical surface (Figure 4a). The enteroendocrine cell marker, Chromogranin A, was expressed by multiple cells concentrated on distinct areas of the culture with clearly thicker extracellular matrix beneath the cells (Figure 4a, arrowhead). The localization of Chromogranin A was concentrated on the apical third of the cells (Figure 4a). The gene expression analyses showed that both substrata induced differentiation of cells, which expressed enterocyte marker genes, *PEPT1* and *villin (*Figure 4b). Laminin increased *villin* gene expression (*p* = 0.001). *CYP3A4* expression was low based on cycle threshold (Ct:) values, which were approximately 30 (data not shown) in all analyzed samples, but expression was higher in iPSC-derived SIECs than in Caco-2 cells. Taken together, both Geltrex and laminin511 substrata supported differentiation and maturation of polarized small intestinal epithelial cells from iPSC-derived posterior definitive endoderm cells.

### 2.5. Functional Characterisation of Produced Small Intestinall Epithelial Cells

Dipeptide uptake, which is performed by SLC15A /PEPT 1 protein activity, is essential for enterocyte functionality [26]. The enterocyte-specific functionality of iPSC-SIECs differentiated and cultured on Geltrex or laminin was studied with dipeptide uptake assay, where β-Ala-Lys-AMCA was used a PEPT1 substrate, and ibuprofen as a PEPT1 inhibitor. The amount of dipeptide uptake was quantified by evaluating the intensity of the fluorescence. The intensity of the fluorescence was comparable in both iPSC-SIEC cultures, but the most prominent inhibition in dipeptide uptake was seen in iPSC-SIECs cultured on Geltrex (*p* = 0.0001), but in iPSC-SIECs cultured on laminin the inhibition was statistically significant (*p* = 0.002). The intensity of the fluorescence in single cells was higher in iPSC-SIECs (Figure 5b,c,e,f) than in Caco-2 cells (Figure 5d,g). The inhibition seen in Caco-2 cells was not statistically significant (*p* = 0.29) (Figure 5a).

Efflux transporters such as multi-drug resistance protein 1 (MDR-1)/P-glycoprotein (P-gp), and multi-drug transport protein (MRP1) are abundantly expressed in intestinal enterocytes [26,27] and serve to protect cells against xenobiotics [22]. Thus, in the final step we studied the efflux pump functionality in the iPSC-SIECs differentiated and cultured on Geltrex or laminin, which was performed with Calcein retention assay. Calcein-AM, a substrate for MDR1/P-gp and MRP1 proteins [28], is colorless and cell permeable until cloven by intracellular esterases. The efflux pump inhibitors (e.g., Cyclosporin A and Verapamil) allow esterases more time to metabolize calcein-AM and thus increase the intracellular fluorescence. In the retention assay incubation with efflux pump inhibitors, Cyclosporin A and Verpamil, increased intracellular fluorescence of single cells in iPSC-SIEC on Geltrex, but as there was high variation within the culture (Figure 6e,h) the overall retention did not differ from that of the control (Figure 6b). In iPSC-SIECs cultured on laminin both Cyclosporin A and Verpamil, increased overall intensity of the fluorescence from 100% to 119% and 113%, respectively, and the inhibition induced by Cyclosporin A was statistically significant (*p* = 0.009). In Caco-2 cultures the untreated cells had high pumping efficiency which is seen as low overall fluorescence (Figure 6d). In Caco-2 cultures especially Cyclosporin A, but also Verapamil induced an increase in fluorescence intensity. Both resulted inhibitions were statistically significant, Cyclosporin A (*p* = 0.0001) and Verapamil (*p* = 0.032).

## 3. Discussion

It is known that animal-derived components induce variation in the experimental outcomes. Therefore, we aimed to refine previously published 2D iPSC-derived small intestinal epithelial differentiation protocols [22,24] by using animal-free components. Previous protocols included two main components that were animal derived: FBS in the cell culture medium, and substrata, either BD Matrigel, or collagen. In the process, xeno-free and chemically defined components were used and their suitability verified. More specifically, the FBS was substituted with KO-SR, and BD Matrigel, here Geltrex, was substituted with recombinant laminin. In addition, we refined the quantities of chemically defined components used in the early differentiation, from iPSC to DE. The differentiation of 2D cultured iPCSs toward SIEC was followed and characterized by qPCR, immunofluorescence imaging, and functionality assays.

An early study by D’Amour et al. showed that 0.5% FBS in differentiation medium in comparison instead of higher concentrations resulted in higher number in DE-cells suggesting that high concentrations of FBS have inhibitory effects on DE-differentiation [29]. Another study reported potentiating effects of FBS by showing that smaller amounts of rhWnt3a are required when 0.5 % FBS is used instead of serum free condition [30]. In our tests we saw dramatic increase in differentiation efficiency when FBS was omitted and replaced with KO-SR. In a recent study by Ghorbani-Dalini et al. reported that cells differentiated in the presence of KO-SR produced more SOX17 + FOXA + cells [25]. This effect was replicated in our data, but we observed a greater change in differentiation efficiency between these conditions. The reason for this effect is suggested to be activin signaling antagonization by active PI3K signaling caused by insulin or insulin-like growth factor (IGF) in FBS [31]. This explanation seems plausible but remains unverified here, as KO-SR and B27 also contain insulin and the exact amounts of these components are proprietary and thus cannot be quantitatively compared to FBS.

Activin-A is the core inducer of DE-differentiation and often combined with Wnt-signaling inducer, usually recombinant Wnt3a or CHIR99021 [19,30,32]. When different combinations of inducers were compared, CHIR was shown to be the superior option as regards economy and efficiency, but at the same time inducing the mesodermal induction [30]. In our study same observations were made, but the difference between 2.5 µM and 5 µM CHIR was smaller than suggested by other authors [19,21,22]. Moreover, we observed that longer exposure of CHIR reduced the number of cells. For early differentiation only 100 ng/mL of Activin A is often used for iPSC to DE induction [18,22]. In our case we saw no SOX17 or FOXA2 expressing cells. It may be that feeder cells underneath iPSCs produce Wnt-inducers which support DE formation better than Geltrex or recombinant laminin which were used in this study. Here, we established a method for efficient DE -differentiation where cells were exposed first 24 h to 2.5 µM CHIR99021 and 100 ng/mL Act-A in the presence of KO-SR, and continued for 48 to 72 h on 100 ng/mL Act-A in the presence of KO-SR.

Two leading 2D iPSC-SIEC research groups used different posteriorizing treatments: the group led by Professor Mizugichi uses LY2090314 [19], and the group led by Professor Matsunaga used bFGF [22]. We were able to posteriorize DE cells with both components, bFGF or LY2090314. However, when the PDE cells were plated on either Geltrex or recombinant laminin coated transwell insets for the SIEC differentiation, the PDE cells failed to adhere and were lost (data not shown). In the paper by Negoro et al. [19], only the induction medium was changed when DE was altered to PDE induction. The procedure of changing only the induction medium is gentler for the cells than the detachment, replating, simultaneously with the change of medium composition, in our protocol. In the end, after several attempts we were finally able to obtain viable LY2090314-induced PDE cells which after SIEC differentiation expressed enterocyte-specific genes (data not shown). Overall, these findings indicate that in our case, SIEC differentiation can be achieved using different biological dispositions as well as by different signaling pathway combinations.

Previous reports by Brafman et al. [33] and Taylor-Weiner et al. [34] have found that fibronectin is a potent definitive endoderm inducer. Later, Rasmussen et al. [35] evaluated 500 different extracellular matrix protein combinations and found that Collagen I was the most potent DE inducer. As we wished to make this method entirely xeno-free and because knew that recombinant collagen is not commonly used [36], we decided to choose laminins. Laminins are heterotrimeric proteins having α-, β-, and a γ-chain, where α is found in five, α in four, and γ in three variants, respectively. Laminin 111 is composed of α1, β1, and γ1 chains [37]. In human intestines, the laminin 111 is reported to reside in the crypt area [38,39] and laminin 511 is mainly found in the differentiated tissues [39]. In previous 2D iPSC-SIEC differentiation protocols the early differentiation form iPSCs to DE is performed on feeder layer [22], or on BD Matrigel [19,24]. The final differentiation from posterior definitive endoderm to SIECs has been performed on BD Matrigel [21,22,24,40]. Here we decided to use laminin 111 in the early differentiation (from iPSC to PDE) and laminin 511 in the final differentiation from PDE to SIEC. As a control we used Geltrex, a liquified form of BD Matrigel, which is a mixture of collagen IV, laminins, and other extracellular matrix components. In the early differentiation from iPSCs to DE, laminin was more potent in producing *SOX17* than Geltrex, and at the same time the expression of pluripotency marker gene *Nanog* was decreased. The numbers of CDX2 and SOX17 double-positive PDE cells were the same on Geltrex and laminin 111 after bFGF induction. We were able to differentiate a low number of viable SOX17-positive cells on Geltrex with LY209314, but on laminin 111 there were few cells left (data not shown). It might be that LY209314 does not support the expression of laminin 111 and laminin 511-binding integrins, as does Geltrex with several laminins, collagen IV, entactin, and heparin sulfate proteoglycans. This remains to be evaluated in future projects.

When the posterior definitive endoderm cells were differentiated toward SIEC both laminin and Geltrex supported the maturation. This culture method supported the best induction of enterocyte type cells determined with PEPT1 expression. There were distinct areas of enteroendocrine type cells with Chromogranin A expression. We also evaluated goblet cell induction by determining mucin protein expression. Mucin positive cells were only found in single, rare spots on Geltrex cultured cells but not on laminin 511 cultured cells (data not shown).

The expression of enterocyte marker genes (*PEPT1* and *villin*) was higher in cells grown on laminin than on Geltrex. In previous functionality assay the PEPT1 dipeptide transported functionality is evaluated with single image. In functionality assays iPSC-SIEC on both substrata were more potent in transporting β-Ala-Lys-AMCA than Caco-2 cells. Thus, the inhibition induced by the ibuprofen was also more apparent. In the dipeptide uptake assay iPSC-SIECs on laminin coating the inhibition was not as drastic as seen in iPSC-SIECs on Geltrex. When the functionality of MDR-1 pumps was assessed in calcein retention assay, the iPSC-SIECs on laminin evinced clearer retention than iPSC-SIECs on Geltrex. Thus, taken together, these findings suggest that small intestinal epithelial cells with typical small intestinal epithelial gene and protein expression and functionality can be generated when iPSCs are differentiated toward SIEC on recombinant laminin substrata.

In our cultures, the enterocyte markers PEPT1 and villin were mainly localized on the apical side of the cells. Given on the localization of these markers [41] we can conclude that both Geltrex and recombinant laminin support polarization of SIECs. To the best of our knowledge, only few 2D iPSC-SIEC papers have provided XZ and YZ scans of iPSC-SIEC immunofluorescence images [19,41,42]. When the morphology of iPSC-SIEC presented in these is compared to the morphology of the cells produced in this study, we can conclude that the cells generated in our study are columnar in shape [19,41,42].

Substrata is known to direct extracellular matrix production. In XZ and YZ scans of iPSC-SIEC immunofluorescence images, we noted that the thickness of extracellular matrix layer after 21 days of culture on laminin 511 was always thinner and more homogenous, than after culture on Geltrex. Geltrex is a basement membrane extracted from murine Engelbreth–Holm–Swarm (EHS) tumors that contain laminin, collagen IV, entactin, and heparin sulfate proteoglycans and traces of growth factors. When compared to recombinant laminin 511 Geltrex is richer and more heterogenous in composition than recombinant laminin 511. Thus, Geltrex can provide more binding sites and other cues for cells and differently induce the production of extracellular matrix than recombinant laminin 511. In this study we only determined how different substrata affects to cellular differentiation. However, determination the biochemical composition of extracellular matrix after 21 days of culture on different substrata is important, and that will be one of the priorities in our future projects.

We started this project using defined, but not xeno-free B27 and N2 supplements. Thus, the method presented here is not an entirely xeno-free 2D SIEC differentiation method. By altering these two components to the already existing, commercial xeno-free B27 and N2 this method can be converted into an entirely xeno-free method. Differentiation efficiency with entirely xeno-free 2D SIEC differentiation method is one of our future aims.

In summary, we refined 2D iPSC-SIEC differentiation toward a xeno-free method. First, our data suggest that differentiation toward DE is more efficient without FBS. Second, both collagen- and laminin-based coating support the differentiation of DE, PDE and SIECs from iPS-cells, but small intestinal epithelial cells differentiated on laminin evinced slightly more enterocyte specific cellular functionality. We can conclude that functional SIEC can be generated without animal-derived substances.

## 4. Materials and Methods

### 4.1. IPSC Culture

The 04602.WT iPSC line, derived in Prof. Katriina Aalto-Setälä’s laboratory (Tampere University, Tampere, Finland), was acquired from adult dermal fibroblasts by viral transduction and characterized as described earlier [43,44]. The 04602.WT line was acquired from a donor with no reported long-term illnesses. During the time of the experiment, the karyotypes of iPSC line was normal (Fimlab Laboratories, Tampere, Finland). iPSCs were maintained on Geltrex (Gibco; Thermo Fisher Scientific, Waltham, MA, USA; diluted 1:100) coated Cell-Bind treated 6-well plates (Corning, Corning, NY, USA) in mTeSR1 medium (StemCell Technologies, Vancouver, BC, Canada) with penicillin–streptomycin (P/S; Gibco, Paisley, UK). When approximately 90% confluency had been reached, the cells were passaged with Versene (Gibco, Grand Island, NY, USA) and seeded at 3–4 × 10^4^ cells/cm^2^.

### 4.2. In Vitro Differentiation to Definitive Endoderm

For the definitive endoderm differentiation, the iPSCs were passaged on one of the following substrata: Geltrex, human recombinant laminin 111 (LN111; Biolamina, Stockholm, Sweden; 2 µg/cm^2^), human fibronectin (FN; Sigma-Aldrich; Merck, Darmstadt, Germany; 5 µg/cm^2^) or a combination of laminin 111 and fibronectin. Cell-Bind-treated 24-well plates were used and cells seeded at 1–1,15 × 10^5^ cells/cm^2^ in mTeSR1 medium and cultured for 24 h. Medium was then changed to RMPI1640 (Gibco, Thermo Fisher Scientific, Paisley, UK) supplemented with GlutaMax (Gibco, Thermo Fisher Scientific, Paisley, UK), 100 ng/mL Activin A (Act-A; Miltenyi Biotec, Bergisch Gladbach, Germany) and P/S. The medium was further supplemented with either FBS (0.5 or 2%) combined with 1 X B27 supplement without Vitamin A (B27; Gibco, Thermo Fisher Scientific, Waltham, MA, USA) or with only B27 for the first 24 h and then 0.2 or 2% KnockOut Serum replacement (KO-SR; Gibco, Thermo Fisher Scientific, Paisley, UK) for the remaining differentiation period. In addition, Wnt-signaling inducers recombinant human Wnt3a (rhWnt3a; R&D Systems, Minneapolis, MN, USA) or CHIR99021 (CHIR; Tocris Biosciences, Bristol, UK) were used at different concentration for the first 24 h or for the total differentiation period of 72 h. Medium was refreshed daily.

### 4.3. In Vitro Differentiation to Posterior Definitive Endoderm

After the definitive endoderm differentiation step, the cells were washed briefly with Dulbecco’s phosphate buffered saline (DPBS; Thermo Fisher Scientific, Paisley, UK) and the medium was changed to High glucose DMEM (Thermo Fisher Scientific, Grand Island, NY, USA) supplemented with non-essential amino acids (NEAA; Thermo Fisher Scientific, Waltham, MA, USA), GlutaMax, 10% KO-SR, P/S. Two combinations of differentiation factors were used: 20 nM LY2090314 (MedChem Express, Monmouth Junction, NJ, USA) or 250 ng/mL bFGF (Miltenyi, Bergisch Gladbach, Germany) as suggested by others [19,21,22]. The posterior definitive endoderm differentiation was carried out on Geltrex and laminin 111 substrata. The differentiation was carried out for four days, and medium was refreshed once, at day two.

### 4.4. In Vitro Differentiation to Small Intestinal Epithelial Cells

Cells priorly differentiated into posterior definitive endoderm were passaged to 0.3 cm^2^ 24-well plate PET TC cell culture inserts with 1 µm pores (Sarstedt, Nümbrecht, Germany) at varying densities. Cells were detached using Versene and passed through 40 µm nylon strainer (Falcon; Corning, Corning, NY, USA). Inserts were coated with human recombinant laminin 511 (LN511; Biolamina, Stockholm, Sweden; 2 µg/cm^2^) for cells originating from LN111 or Geltrex surfaces and with Geltrex for cells originating from Geltrex surface. Medium used included 1:1 mixture of High Glucose DMEM and F-12 supplement (Gibco, Thermo Fisher Scientific, Paisley, UK) with 20% KO-SR, NEAA, GlutaMax, B27 (-Vit A), N2 (Thermo Fisher Scientific, Paisley, UK) and P/S. The medium was supplemented just prior to replenishment with 30 ng/ ml rhWnt3A (R&D Systems), 30 ng/mL rhR-spondin-3 (Miltenyi Biotech), 30 ng/mL rhNoggin (Miltenyi Biotech, Bergisch Gladbach, Germany), 30 ng/mL rhEGF (R&D Systems, Minneapolis, MN, USA), 20ng/mL rhIGF (R&D Systems, Minneapolis, MN, USA), 10 µM SB202190 (Sigma-Aldrich), 1 µM Dexamethasone (Tocris Biosciences), adopted and modified from previous work by other authors [22,24]. Rho-kinase inhibitor Y-27632 (Sigma-Aldrich, Darmstadt, Germany) was included in the medium for the first 24 h. The medium was changed every three days. This differentiation step was carried out for 21 days.

### 4.5. Caco-2 Culture

Caco-2 cells (ATCC, HTB-37) were cultured with minimal essential medium (MEM, Gibco, Thermo Fisher Scientific) supplemented with 10% FBS, 1% NEAA, GlutaMAX, and P/S similarly as in [19]. For differentiation of Caco-2 cells, Caco-2 cells plated on Geltrex coated inserts at a density of 100,000 cells/cm^2^. Medium was replenished every second day. Cells were cultured for ten days. At the end of culture, transepithelial resistance was measured with Millicell-ERS volt-ohm meter (Millipore Corporate, Darmstadt, Germany), and Caco-2 cultures were regarded as confluent when resistance exceeded 600 ohms/cm^2^.

### 4.6. Immunofluorescence Staining and Imaging

Cells were washed three times with DPBS and fixed with 4% paraformaldehyde (Acros Organics; Thermo Fisher Scientific, Waltham, MA, USA) for ten minutes, permeabilized with 0.1% Triton X-100 for ten minutes and blocked in 3% bovine serum albumin for one hour. Primary antibodies (Table 1) were diluted in 0.5% BSA and incubated with the sample for one hour followed by treatment with secondary antibodies (Table 1) in a parallel fashion. Cells were washed three times after every step, except after the blocking. Samples were mounted using Vectashield with DAPI (Vector Laboratories, Burlingame, CA, USA). All steps were conducted at room temperature.

IPSC, DE, PDE were stained and imaged in their culture vessels. Imaging of DE samples was conducted using Olympus IX-51 epifluorescence microscope (Olympus Corporation, Tokyo, Japan), and images of PDE samples on different substrata (Figure 3b) with the same microscope Olympus IX-51, but with new Hamamatsu Orca Flash4.0LT+ *sCMOS* (Hamamatsu Photonics Europe GmbH, Herrsching am Ammersee, Germany) camera. Images were colorized using Photoshop (Adobe Inc., San José, CA, USA). SILC samples cultured on inserts, were released from the housing by scalpel and the membrane dissected into smaller pieces. Staining was conducted on top of objective glasses according to the method described above. Samples were imaged using LSM780 laser scanning confocal microscope (Carl Zeiss Ag, Oberkochen, Germany) Zeiss Plan Apo 40×/1.4 oil immersion objective, with Diode laser and 405 nm filter, Multiline Argon laser and 488 nm filter, and InTune-tunable pulsed laser and 568 nm filter, pixel size 2048 × 2048. Images were processed with ZEN Black and Zen Blue software (Zeiss).

### 4.7. qRT-PCR

RNA was extracted from the cells after every differentiation step using RNeasy Plus Mini Kit (Qiagen, Hilden, Germany) according to manufacturer protocol. Reverse transcription was carried out using High-Capacity cDNA Reverse Transcription Kit (Applied Biosystems; Thermo Fisher Scientific, Waltham, MA, USA) and Eppendorf MasterCycler thermal cycler (Eppendorf, Hamburg, Germany). The maturation status of cultures after DE, PDE, or SILC induction was assessed by harvesting and analyzing the cells the expression of NANOG (Hs02387400_g1), *T/BXT* (*BRACHYURY* (Hs00610078_m1)), *SOX17* (Hs06634937_s1), *POUF* (OCT4 Hs00999632_g1) mRNA after DE induction, and *SLC15A* (*PEPT1* (Hs00192639_m1)), *Villin* (VIL1 (Hs01031724_m1)), and *CYP3A4* (Hs00604506_m1) mRNA expression after SILC induction with TaqMan^®^ gene expression assays (Applied Biosystems, Inc., Foster City, CA, USA) using FAM labels. Abovementioned target genes were analyzed and compared against the *glyceraldehyde 3-phosphate dehydrogenase* (*GAPDH*; Hs99999905_m1), which was used as an endogenous control. Samples and template-less controls of *NANOG, T/ BXT (BRACHYURY), SOX17, POUF (OCT4), PEPT1, VIL1 and CYP3A4* and *GAPDH* were run in triplicate using the 7300 Real-Time PCR system (Applied Biosystems, Inc. Inc., Foster City, CA, USA) with the following program: 2 min at 50 °C, 10 min at 95 °C; 40 cycles of 15 s at 95 °C, and finally 1 min at 60 °C. Results were analyzed using 7300 System SDS Software 2.4 (Applied Biosystems, Inc., Foster City, CA, USA). The relative quantification of each gene was calculated using Ct values and the 2−ΔΔCt method [45] using *GAPDH* as a calibrator.

### 4.8. Functionality Assays

#### 4.8.1. Dipeptide Uptake Assay

Dipeptide uptake was performed by SLC15A /PEPT 1 protein activity [26], was performed according to the previous protocols [46,47]. Briefly: The cells were either left untreated or pretreated with SLC15A /PEPT 1 inhibiting Ibuprofen (EMD Millipore Corporation, Merck KGaA, Darmstadt, Germany) for 1 h 3 mM as previously described [46,47]. Thereafter, cells were incubated with 25 µM β-Ala-Lys (β-Ala-Lys-AMCA), a fluorescence-labelled tripeptide substrate of PEPT1 (Peptide institute, Osaka, Japan) for 4 h in an incubator, as previously described in [17,19]. Both in the preincubation and the uptake assay Hanks’ balanced salt solution (HBSS, Thermo Fisher, Gibco, Paisley, UK) was used as transport buffer. After the uptake period, cells were washed three times with 1× DPBS, fixed with 4% PFA, and mounted on Vectachield with DAPI. The degree of fluorescence was analyzed by capturing images of randomly selected locations on cell culture inserts. In all experiments the light exposure settings and illumination were maintained constant between samples. In a previous functionality assays PEPT1 dipeptide transport functionality was evaluated from single images from control and inhibited samples [17,19,48]. We wished to have a numerical value from the assay, thus we performed the image analysis from the experiment: Five images from both inhibited and control samples from three technical replicates and three independent differentiation experiments were analyzed. These analyzed images were taken with 20× objective with Olympus BX60 immunofluorescence microscope (Olympus, Hamburg, Germany), Olympus ColorView II camera. The intensity of fluorescence was quantified with Fiji-Image J (Image Processing and Analysis Software, https://imagej.net/software/fiji/, downloaded 17th of October 2019). The data from different sample series (iPSC-SIEC on Geltrex, iPSC-SIEC on laminin, or Caco-2 on Geltrex) were normalized against the average of untreated samples.

#### 4.8.2. Calcein AM Extrusion Assay

The functionality of efflux transporters, multi-drug resistance protein 1 (MDR-1)/ P-glycoprotein (P-gp) was assessed in calcein retention assay using calcein–acetoxymethyl (AM) which is a substrate for MDR1 [28]. The experiment was performed as previously described [3,27]. Briefly, the cells were pre-equilibrated with 25 mM HEPES- buffered Hank’s balanced salt solution in presence or absence of MDR-1 inhibitor, 15 mM cyclosporin A (EMD Millipore Corporation, Merck KGaA, Darmstadt, Germany) or 500 uM Verapamil (EMD Millipore Corporation, Merck KGaA, Darmstadt, Germany), for 20 min at 37 °C. Thereafter, calcein-AM (EMD Millipore Corporation, Merck KGaA, Darmstadt, Germany) was added to a final concentration of 2 mM and incubation continued for 20 min at 37 °C. Thereafter cells were washed with ice cold HBSS, taken on ice, and visualized immediately 20× objective with Olympus IX51 immunofluorescence microscope and images captured with Hamamatsu Orca Flash4.0LT+ sCMOS camera (Hamamatsu Photonics Europe GmbH). The degree of fluorescence was analyzed by capturing images of randomly selected locations on cell culture inserts. In all experiments the light exposure settings and illumination were maintained constant between the samples. Five images both from treated and untreated, i.e., control samples from three technical replicates and 3–4 independent differentiation experiments were analyzed. The intensity of fluorescence was quantified with Fiji-Image J 64 bit (Image Processing and Analysis Software https://imagej.net/software/fiji/, downloaded 17th of October 2019). The data from different sample series (iPSC-SIEC on Geltrex, iPSC-SIEC on laminin, or Caco-2 on Geltrex) were normalized against the average of untreated samples.

### 4.9. Statistical Significance

The statistical significance of numerical data were analyzed using IBM SPSS Statistics version 26, with a two tailed Mann–Whitney U test. The number of biological and technical replicates is indicated in the figure legends.

### 4.10. Ethical Issues

The study was approved by the Ethics Committee of Pirkanmaa Hospital District to derive and expand iPSC lines (R08070 and R12123), and written consent was obtained from all fibroblast donors. All the patients were over 18 years old. The use of these cell lines for gastrointestinal research was approved by the Ethical Committee of Pirkanmaa Hospital District (R20008). No new cell lines were derived in this study.

## 5. Conclusions

Our data suggest that differentiation toward a definitive endoderm is more efficient without FBS. Both collagen- and laminin-based coating support differentiation of definitive endoderm, posterior definitive endoderm, and small intestinal epithelial cells from iPS-cells, but small intestinal epithelial cells differentiated on laminin evinced slightly more enterocyte specific cellular functionality. Here, we showed that functional small intestinal epithelial cells can be generated with animal-free medium and substrata, i.e., without serum and on recombinant laminin.

## Figures and Tables

**Figure 1 ijms-23-01312-f001:**
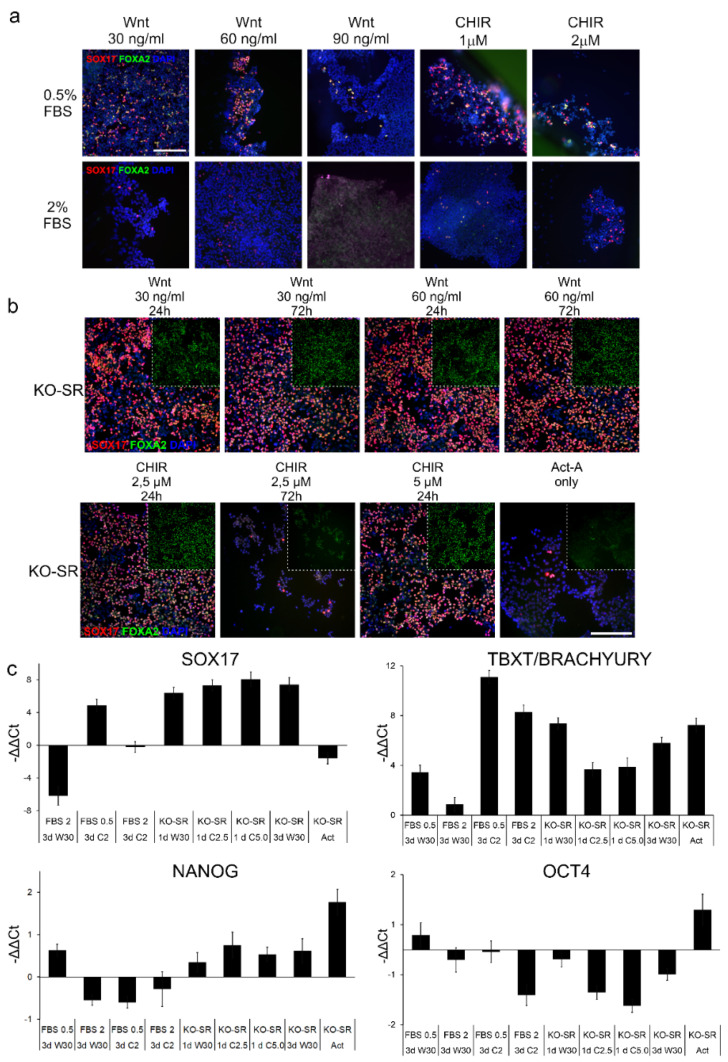
Effect of culture conditions in generation definitive endoderm. (**a**) iPSCs were differentiated to definitive endoderm by exposing the cells to two different concentrations of fetal bovine serum (FBS) 0.5% and 2%, different concentrations of rhWNT3a (Wnt) or CHIR99021 (CHIR) for 72 h All other supplements were held constant. Cells were stained with endodermal markers, SOX17 (red) and FOXA2 (green). Number of biological replicates *n* = 3, number of technical replicates *n* = 2. Scalebar 200 µm (**b**) Effect of different exposure times and concentrations of Wnt3A or CHIR in generation of definitive endoderm: iPSCs were differentiated to definitive endoderm by exposing the cells to different concentrations of rhWNT3a (Wnt) or CHIR99021 (CHIR) for 24 or 27 h. All these experiments were performed in absence of FBS, and presence of knock-out serum replacement (KO-SR). All other supplements were held constant. Cells were stained with endodermal markers, SOX17 (red) and FOXA2 (green). Number of biological replicates *n* = 3, number of technical replicates *n* = 2. Scalebar 200 µm (**c**) Effect of different culture conditions on the expression of endoderm marker *SOX17*, mesoderm maker *TBXT* and pluripotency markers, *NANOG* and *OCT4*. IPSCs differentiated to definitive endoderm with 100 ng/mL Act-A and 2.5/5 µM CHIR99021 (C) or 30 ng/mL WNT3a (W). CHIR (C) and WNT3a (W) were introduced to the differentiation media for one or three days (1d/3d) The medium was supplemented with 0.5 or 2% FBS with B27VitA (FBS) or B27–Vit A followed by 0.2% and 2% knockout serum replacement medium (KO-SR) for three days, respectively. Control cells were exposed to neither CHIR (C) nor WNT3a (W) but only to Activin-A (Act). *SOX17* expression was calibrated against a condition where cells were induced to differentiate with 100 ng/mL Act-A + 30 ng/mL WNT3a (W) (1d) and medium supplemented with FBS. Data expressed as mean ± SD.

**Figure 2 ijms-23-01312-f002:**
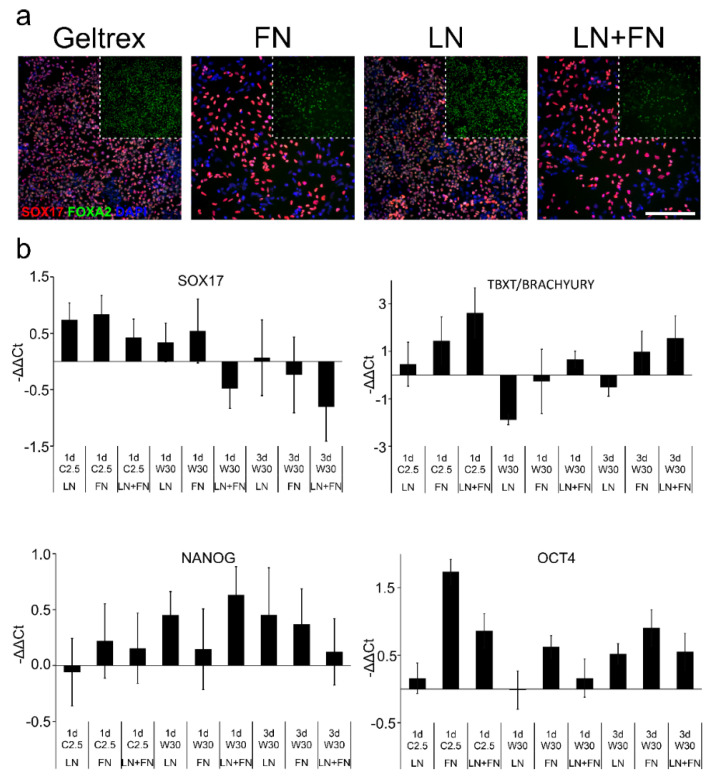
(**a**) Expression of endodermal marker proteins SOX17 (red) and FOXA2 (green) after differentiation of iPSCs to definitive endoderm (DE) on cell culture substrata: Geltrex, Fibronectin (FN), Laminin111 (LN) or LN+FN. Differentiation toward DE was induced in the presence of 100 ng/mL Act-A and 2.5 µM CHIR99021 and KO-SR. Number of biological replicates *n* = 5, number of technical replicates *n* = 2. Scalebar 200 µm. (**b**) Gene expression of endoderm marker *SOX17*, mesoderm maker *TBXT* and pluripotency markers *NANOG* and *OCT4* in iPSCs differentiated to definitive endoderm with 100 ng/mL Act-A and 2.5 µM CHIR99021 (C) (1d) or 30 ng/mL WNT3a (W) (1d/3d). Expression of genes was calibrated against cells differentiated with the same inducers on Geltrex coated surfaces (Geltrex result, which is nor shown in graph = 0). Data expressed as mean ± SD.

**Figure 3 ijms-23-01312-f003:**
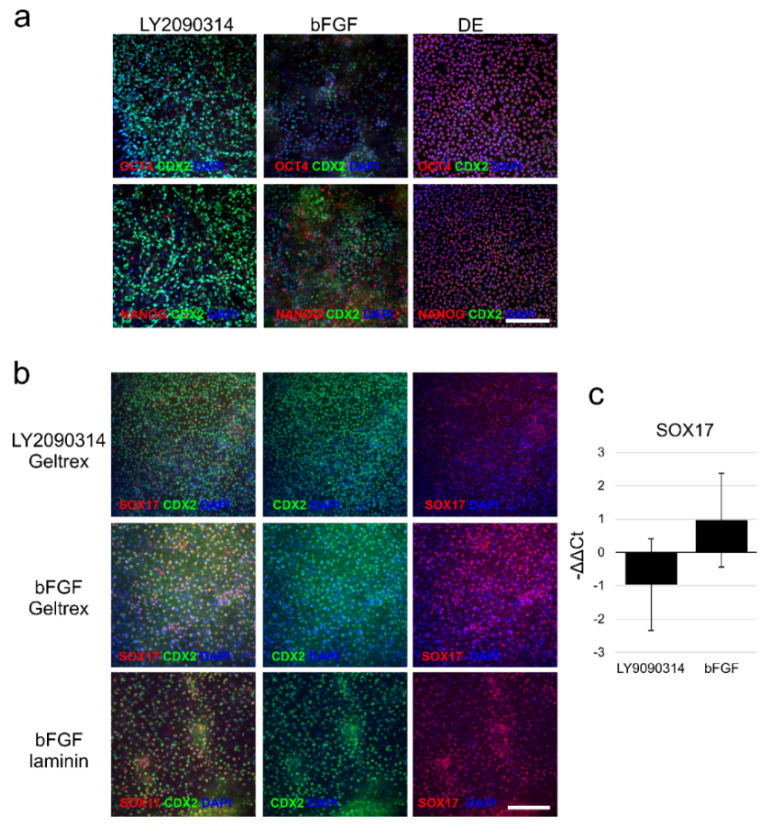
Differentiation toward posterior definitive endoderm. (**a**) Expression of posterior definitive endoderm (DE) marker CDX2 (green), and pluripotency markers OCT4 and NANOG (red). Cells were exposed to LY2090314 or bFGF and induced to differentiate toward PDE on Geltrex. As a control, DE cells were stained against the same antibodies. In the DE there was no CDX2 stain but a clear nuclear localization of pluripotency markers. Scalebar 200 µm. Number of biological replicates *n* = 5, number of technical replicates *n* = 2. (**b**) Expression of posterior definitive endoderm (DE) markers CDX2 (green) and SOX17 (red). Cells were exposed to LY2090314 or bFGF and induced to differentiate toward PDE either on Geltrex or on laminin 111. Number of biological replicates *n* = 2, number of technical replicates *n* = 3. (**c**) In quantitative RT-PCR the bFGF induced higher SOX17 expression than LY2090314. Data expressed as mean ± SD.

**Figure 4 ijms-23-01312-f004:**
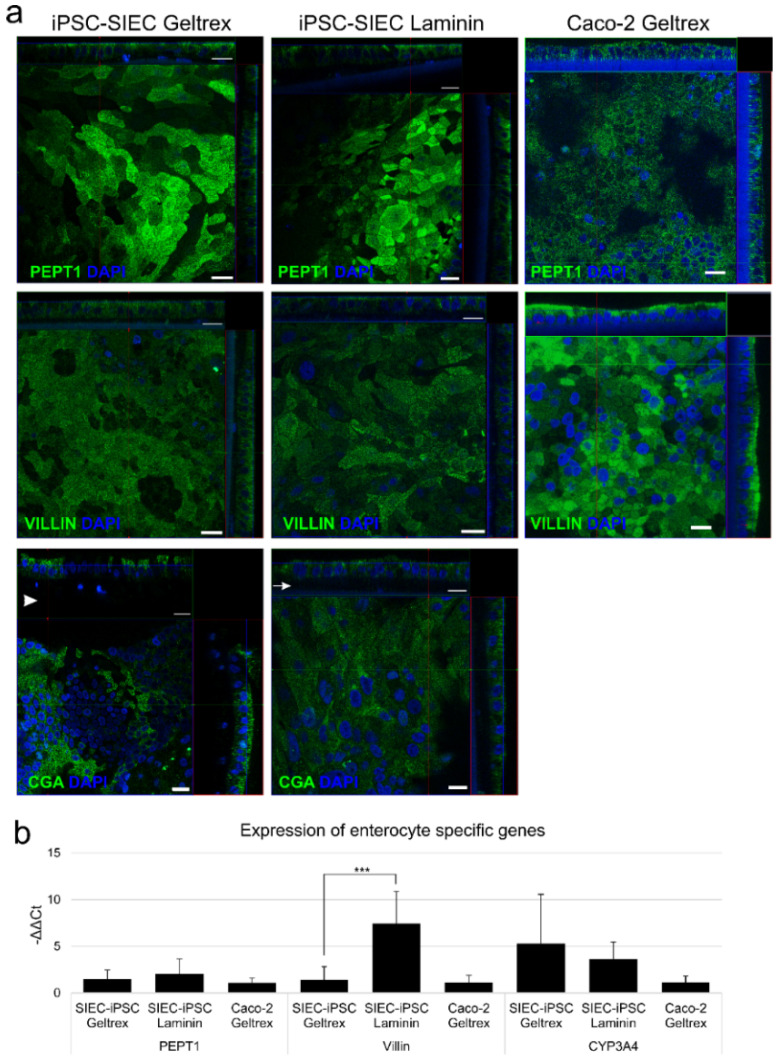
Differentiation toward small intestinal epithelium (**a**) Intracellular localization of small intestinal epithelial cell markers in SIEC-iPSCs cultured either on Geltrex or laminin 511. Caco-2 cells grown on Geltrex served as a control. Enterocyte marker proteins PEPT1 (green) and villin (green), and enteroendocrine cell marker chromogranin A (CGA, green) were visualized with indirect immunofluorescence and confocal microscopy. In samples stained with Chromogranin A, the arrowhead in Geltrex, and arrow in laminin iPSC-SIEC samples point to extracellular matrix. Number of biological replicates *n* = 4, number of technical replicates *n* = 2. Scalebar 20 µm. (**b**) The expression of enterocyte marker genes *PEPT1, villin* and *CYP3A4* was analyzed with qRT-PCR. The expression of iPSC-SIECs were calibrated against iPSC-SIEC cultured on Geltrex. Number of biological replicates *n* = 3, number of technical replicates in every biological replicate *n* = 3. Data expressed as mean ± SD. The statistical significance: *** (*p* ≤ 0.001).

**Figure 5 ijms-23-01312-f005:**
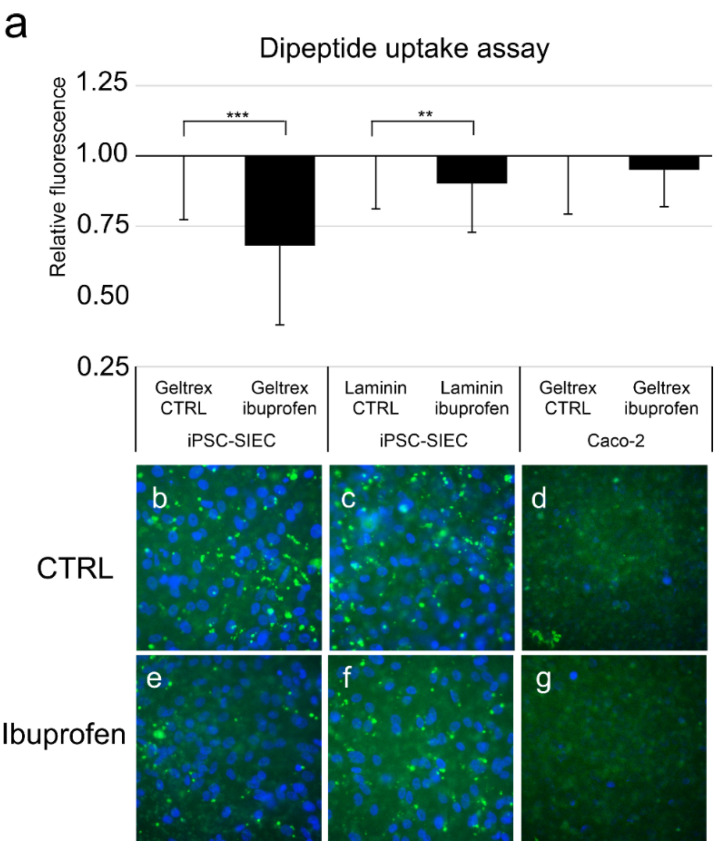
Small intestinal epithelial cell specific functionality assessed with dipeptide uptake assay. Posterior definitive endoderm differentiated iPSCs were cultured on Geltrex or Laminin511 and differentiated toward small intestinal epithelial cells. Caco-2 cells cultured on Geltrex served as control. (**a**) The intensity of fluorescence analyzed from micrographs after dipeptide (D-Ala-Leu-Lys-AMCA) uptake analyses +/− ibuprofen treatment. (**b**) iPSC-SIEC cultured on Geltrex, CTRL i.e., without ibuprofen, (**c**) iPSC-SIEC cultured on laminin, CTRL, i.e., without ibuprofen, (**d**) Caco-2 cultured on Geltrex, CTRL, i.e., without ibuprofen, (**e**) iPSC-SIEC cultured on Geltrex, with ibuprofen, (**f**) iPSC-SIEC cultured on laminin, with ibuprofen, (**g**) Caco-2 cultured on Geltrex, with ibuprofen. Number of biological replicates *n* = 3–4, number of technical replicates *n* = 3. Data expressed as mean ± SD. The statistical significance: ** (*p* ≤ 0.005) and *** (*p* ≤ 0.001) indicating the significance between the indicated samples.

**Figure 6 ijms-23-01312-f006:**
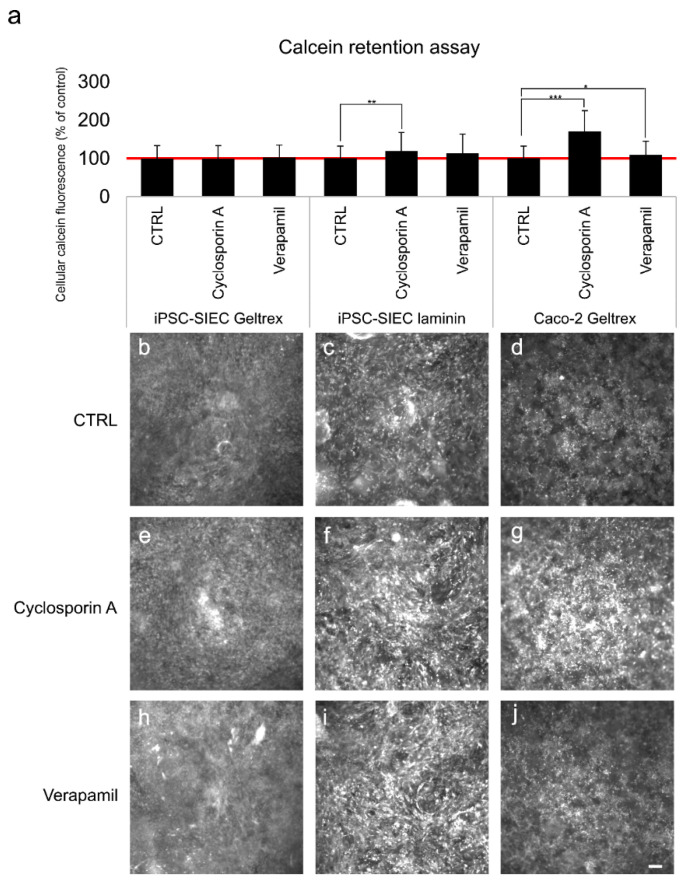
The functionality of efflux transport proteins. The functionality of efflux transporters was assessed with calcein retention assay in the absence (=CTRL) or presence of efflux protein inhibitors Cyclosporin A or verapamil from the small intestinal epithelial cells differentiated from posterior definitive endoderm on Geltrex or laminin511. (**a**) Functionality, which is seen as retention, is expressed as a percentage of fluorescence relative to the control (control = 100%, is marked with the red line. (**b**–**j**) micrographs taken to quantitate retention efficiency (**b**) iPSC-SIEC cultured on Geltrex, without inhibitors (CTRL), (**c**) iPSC-SIEC cultured on laminin, without inhibitors (CTRL), (**d**) Caco-2 cultured on Geltrex, without inhibitors (CTRL), (**e**) iPSC-SIEC cultured on Geltrex, with Cyclosporin A, (**f**) iPSC-SIEC cultured on laminin, with Cyclosporin A, (**g**) Caco-2 cultured on Geltrex, with Cyclosporin A. (**h**) iPSC-SIEC cultured on Geltrex, with Verapamil, (**i**) iPSC-SIEC cultured on laminin, with Verapamil, (**j**) Caco-2 cultured on Geltrex, with Verapamil. Scalebar 100 µm. Number of biological replicates *n* = 3–4, number of technical replicates *n* = 3. Data expressed as mean ± SD. The statistical significance: * (*p* ≤ 0.05), ** (*p* ≤ 0.005) and *** (*p* ≤ 0.001) indicating the significance between the indicated samples.

**Table 1 ijms-23-01312-t001:** Antibodies used for immunofluorescence staining. PDE: posterior definitive endoderm; Abcam: Cambridge, UK; EMD Millipore, Merck, Darmstadt, Germany; Novus Biologicals, Centennial, CO, USA.

**Primary Antibodies**					
**Target**	**Target marker for**	**Manufacturer**	**Catalog**	**Host**	**Dilution**
Oct4	pluripotency	R&D	AF1759	goat	1:300
Nanog	pluripotency	R&D	AF1997	goat	1:300
Sox17	endoderm	R&D	AF1924	goat	1:300
Cdx2	posterior definitive endoderm	Abcam	ab76541	rabbit	1:300
Foxa2	endoderm	EMD Millipore	07-633	rabbit	1:300
Villin	brush border	Abcam	ab130751	rabbit	1:200
Pept1	enterocytes	Novus Biologicals	NBPI-92005	rabbit	1:100
Chromogranin A	enteroendocrine cells	Abcam	ab15160	rabbit	1:100
**Secondary Antibodies**					
**Target**	**Manufacturer**	**Wavelength**	**Cat**	**Host**	**dilution**
Mouse IgG	Life Technologies	568	A11031	Goat	1:500
Goat IgG	Invitrogen	568	A11057	Donkey	1:1000
Rabbit IgG	Invitrogen	488	A21206	Donkey	1:500

## Data Availability

The detailed data of the current study are available from the corresponding authors on reasonable requests.

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
