# Peer review of "Toward Xeno-Free Differentiation of Human Induced Pluripotent Stem Cell-Derived Small Intestinal Epithelial Cells"

_ijms, 2022, doi:10.3390/ijms23031312_

Round 1

Reviewer 1 Report

This paper investigates the possibility of refinement for 2D iPSC-SIEC differentiation by using a xeno-free method to avoid animal-derived substances. The authors suggest that differentiation towards DE is more efficient without animal-derived FBS and that small intestinal epithelial cells differentiated on laminin showed slightly more enterocyte specific cellular functionality.

The results of the study are quite interesting and may be important, but there are major issues that need to be addressed before publication.

lines 59-60

"2D culture is accessible from both sides of the cells, i.e. from both sides of the culture, and thus advantageous in high-throughput pharmacokinetic testing."

"both sides of the cells" and " both sides of the culture" are confusing.

I would suggest the authors to explain and clarify them.

line 94

Explain the role of Activin A in culture on DE-differentiation.

More detailed explanation of Activin A is needed here.

line 137

"2.2. Effects of Substrata on the Differentiationtowards Definitive Endoderm"

A space is required in this subtitle, between "Differentiation" and "towards".

lines 157-158

"Regardless of the coatings, in the Wnt3a condition the expression of TBXT/BRACHYURY expression decreased."

I cannot understand the sentence. Isn't there discrepancy between the sentence and Figure 2b TBXT/BRACHYURY?

lines 164-166

"As fibronectin alone and laminin and fibronectin coatings resulted in a lower yield of SOX17 and FOXA2 positive cells,"

Authors should explain the possible discrepancy between the results of Figure 2a and Figure 2b. I cannot see the lower yield of SOX17 and FOXA2 in Figure 2a.

line 182

Explain the role of LY2090314 in culture.

More detailed explanation of LY2090314 is needed here.

lines 218-221

"the thickness of the subepithelial extracellular matrix was greater on Geltrex than on laminin (Figure 4a, arrowhead in Chromogranin on Geltrex culture, and arrow in the same but on laminin)."

The extracellular matrix has been determined by only the subepithelial space. Biochemical determination for extracellular matrix should be needed.

Figure 4a indicates the differentiation of 2D cultured iPCSs towards SIEC, but the results contain the 3D elements by using XZ and YZ scans of immunofluorescence images. Thus, the exact analysis of extracellular matrix of the epithelial cells is very important.

line 247

What is difference between SIEC-iPSCs and iPSC-SIEC?

line 262

"iPSC derived small intestinal cells" had better be described as iPSC-SIEC.

lines 372-373

I would suggest the authors to explain the difference between laminin111 and laminin511.

And show the content of both laminin111 and laminin511 in Geltrex as a control.

Author Response

Reviewer #1

  • lines 59-60: "2D culture is accessible from both sides of the cells, i.e. from both sides of the culture, and thus advantageous in high-throughput pharmacokinetic testing."."both sides of the cells" and " both sides of the culture" are confusing.I would suggest the authors to explain and clarify them.

Our response: It is true that difference was not explicitly explained. Now the explanation starts already on the row 53-54 when the structure of organoids is clarified “…which are closed structures having apical, food ingesting side inside the lumen. Currently organoids...”, and continues in the 2D-culture on row 60-61 the explanation:” In 2D culture, the cells are grown on flat porous surface, which allow handling of culture both from apical and basal side of the cells. As..” is added.

  • line 94: Explain the role of Activin A in culture on DE-differentiation. More detailed explanation of Activin A is needed here.

Our response: We thank for this comment. We have now added clarifying text “…a known TGFβ and nodal signaling inducer [25], was used alone as an inducer…” on the row 96 in the final form of the manuscript.

  • line 137: "2.2. Effects of Substrata on the Differentiationtowards Definitive Endoderm"A space is required in this subtitle, between "Differentiation" and "towards".

Our response: Thank you for the acuity. Space now added between the word and can be found from the row 139 in the final form of the manuscript.

  • lines 157-158: "Regardless of the coatings, in the Wnt3a condition the expression of TBXT/BRACHYURY expression decreased." I cannot understand the sentence. Isn't there discrepancy between the sentence and Figure 2b TBXT/BRACHYURY?

Our response: Thank for this comment, we should have indicated that we meant the coatings which were not mixed. Now that section is corrected to form “ Regardless of the coatings, In the Wnt3a condition the expression of TBXT/BRACHYURY expression decreased in cells cultured on pure laminin 111 or fibronectin, but increased in cells cultured on laminin 111-fibronectin mixture. “. This can be found from the rows 159- 161 in the final form of manuscript.

  • lines 164-166: "As fibronectin alone and laminin and fibronectin coatings resulted in a lower yield of SOX17 and FOXA2 positive cells," Authors should explain the possible discrepancy between the results of Figure 2a and Figure 2b. I cannot see the lower yield of SOX17 and FOXA2 in Figure 2a.

Our response: This is an important note. Cells grown on Geltrex have more intense FOXA2 staining than cells grown on the fibronectin or laminin 111-fibronectin coating. The percentage of SOX17 positive cells is higher in cells grown on Geltrex than in cells grown on laminin 111, fibronectin or laminin 111 – fibronectin mixture. Now the text is corrected to form “Cells grown on Geltrex had more intense FOXA2 staining than cells grown on fibronectin alone or on laminin 111 - fibronectin mixture (Figure 2a). The percentage of SOX17 positive cells, was markedly lower in cells grown on fibronectin alone and on laminin 111- fibronectin mixture than on Geltrex (Figure 2a). As immunofluorescence data generated with the 2.5 µM CHIR and 3d Wnt3A, the immunofluorescence data (Figure 2a) is in line with the gene expression data (Figure 2b). As fibronectin and laminin 111- fibronectin coatings resulted low SOX17 expression…”. This can be found on rows 168 - 175 in the final form of the manuscript.

  • line 182: Explain the role of LY2090314 in culture. More detailed explanation of LY2090314 is needed here.

Our response: It is true that functionality of LY2090314 was not explained. Now the LY2090314 function is explained in more detail. On rows 191-196 in the final form of the manuscript :”LY2090314 is a known glycogen synthase kinase-3 inhibitor, which can activate Wnt/β-catenin signaling [19]. LY2090314 is also shown to promote CDX2 expression, which is a marker of posterior definitive endoderm [19]. “

  • lines 218-221: "the thickness of the subepithelial extracellular matrix was greater on Geltrex than on laminin (Figure 4a, arrowhead in Chromogranin on Geltrex culture, and arrow in the same but on laminin)."

The extracellular matrix has been determined by only the subepithelial space. Biochemical determination for extracellular matrix should be needed.

Figure 4a indicates the differentiation of 2D cultured iPCSs towards SIEC, but the results contain the 3D elements by using XZ and YZ scans of immunofluorescence images. Thus, the exact analysis of extracellular matrix of the epithelial cells is very important.

Our response: We thank for this comment. It is true that notion is only very descriptive.

When cells were grown on laminin (from iPSC to iPSC-DE and iPSC-PDE on laminin 111, and from PDE to iPSC-SIEC on laminin 511), the layer of extracellular matrix was always rather thin, whereas after culture on Geltrex the thickness could vary a lot, which can also seen in Figure 4a. Sentence is now rewritten: “the thickness of the subepithelial extracellular matrix varied more was greater on Geltrex, resulting to some areas thicker extracellular matrix...”. That can be found from the final manuscript on rows 230-231.

The reviewer is completely right. Matrix composition, and determination of biochemical composition of matrix composition is important. In this paper we focused on to evaluate how xeno-free protein coating (laminin) is able to support intestinal epithelial cell differentiation compared to commonly used non-xeno-free (Geltrex) coating. Therefore, in our opinion determination of the resulted matrix composition is out of scope of this paper. In addition, determination of the biochemical composition with new cell cultures, and Mass Spectrometry would take at least six months.

It is true that Geltrex, which is from extracted from murine Engelbreth-Holm-Swarm (EHS) tumors is richer and can differently direct the production of extracellular matrix. We have now added the text to the discussion: “ Substrata is known to direct extracellular matrix production. In XZ and YZ scans of iPSC-SIEC immunofluorescence images we noted that the thickness of extracellular matrix layer after 21 days of culture on laminin511 was always thinner and more homogenous, than after culture on Geltrex. Geltrex, is a basement membrane extracted from murine Engelbreth-Holm-Swarm (EHS) tumors, that contain laminin, collagen IV, entactin, and heparin sulfate proteoglycans and traces of growth factors. When compared to recombinant laminin 511. Geltrex is richer and more heterogenous in composition than recombinant laminin 511. Thus, Geltrex can provide more binding sites and other cues for cells and differently induce the production of extracellular matrix than recombinant laminin 511. In this study we only determined how different substrata affects to cellular differentiation. However, determination the biochemical composition of extracellular matrix after 21 days of culture on different substrata is important, and that will be one of the priorities in our future projects.” This can be found from the rows 424 - 436 in the final form of the manuscript.  

  • line 247: What is difference between SIEC-iPSCs and iPSC-SIEC?

Our response: Thank you for the perceptiveness. This was a remnant of the previous naming. Now that is corrected to the form iPSC-SIECs. Correction can be found in the final form of the manuscript on row 259.

  • line 262: "iPSC derived small intestinal cells" had better be described as iPSC-SIEC.

Our response: We thank for this comment. Now text corrected to iPSC-SIEC. In the final form of the manuscript on row 272.

  • lines 372-373: I would suggest the authors to explain the difference between laminin111 and laminin511. And show the content of both laminin111 and laminin511 in Geltrex as a control

Our response: It is true that the difference between laminin 111 and laminin 511 was not explained. We have now clarified this by adding the text “Laminins are heterotrimeric proteins having α-, β-, and a γ-chain, where α is found in five, α in four, and γ in three variants, respectively. Laminin111 is composed of α1, β1, and γ1 chains [38].” This can be found in the final form of the manuscript on the rows 378 - 380.

Reviewer 2 Report

Saari and colleagues were evaluated the differentiation ability of small intestinal epithelial differentiation of iPSC in 2D culture without animal-derived components.

Authors found that differentiation towards a definitive endoderm is more efficient without FBS and small intestinal epithelial cells differentiated on laminin showed more enterocyte specific cellular functionality.

Major point

  1. iPSC can differentiate into all components of small intestinal differentiated cells, including absorptive cells (enterocyte) as well as secretory cells (Paneth cells, goblet cells, and enteroendocrine cells). This study was focused on the enterocyte differentiation and its functional assay. If possible, authors should compare the expression level of differentiation marker of Paneth cells, goblet cells, and enteroendocrine cells.

  1. Furthermore, iPSC may have a potential capacity of differentiation into stromal cells, such as myofibroblast. In condition with laminin- and collagen-based matrix,

I wonder that iPSC can differentiate into myofibroblasts.

Minor point

1. abstract line 12: two-dimensional 2D -> two-dimensional (2D)

2. page 8, 232: Ct:s, Please make it complete.

Author Response

Reviewer #2

Major point:

  • iPSC can differentiate into all components of small intestinal differentiated cells, including absorptive cells (enterocyte) as well as secretory cells (Paneth cells, goblet cells, and enteroendocrine cells). This study was focused on the enterocyte differentiation and its functional assay. If possible, authors should compare the expression level of differentiation marker of Paneth cells, goblet cells, and enteroendocrine cells.

Our response: This is very important comment. It is already known that when the Wnt induction is low and notch is high at first, and then followed by high bone morphogenic protein induction, the resulted cell type is absorptive progenitor cell. Whereas when Wnt is low and Notch is off in the beginning, and then the a) the Wnt is low, Notch off, IL-4 and IL13 present the resulted cell type is Goblet cell, b) the Wnt is high, Notch off, and FGF present, the resulted cell type is Paneth cell, c) the Wnt is low, Notch off and EGF not present the resulted cell type is enteroendocrine cell (see e.g. Gehart & Clevers, 2019 doi.org/10.1038/ s41575-018-0081-y).

With the method presented in the manuscript we saw emergence of enteroendocrine cells (Chromogranin A positive cells). Those were concentrated on distinct areas in the culture. That is also indicated in the manuscript (see rows 241-244 in the final form of the manuscript). Those areas were rather scattered so that the percentage of enteroendocrine positive cells from the entire culture is rather difficult to determine without dissociation of cells and without FACS. We did see some scattered goblet type cells with mucin staining in cultures on Geltrex, but not on laminin. Those places with mucin positivity, the epithelium was rather flat, and did not express PEPT1 in adjacent cells. With different composition of medium which resembled some extent that used by Kondo et al 2020 (doi:10.1242/bio.049064), we got only mucin positive cells but not enterocytes. That research line was discontinued.

We have now added text about different cell types to the discussion:“ This culture method supported best induction of enterocyte type cells shown with PEPT1 expression. There were distinct areas of enteroendocrine type cells with Chromogranin A expression. We also evaluated goblet cell induction by determining mucin protein expression. Mucin positive cells were only found in single, rare spots on Geltrex cultured cells, but not on laminin511 cultured cells (data not shown).”. This can be found from the rows 398 - 403 in the final form of the manuscript.

  • Furthermore, iPSC may have a potential capacity of differentiation into stromal cells, such as myofibroblast. In condition with laminin- and collagen-based matrix, I wonder that iPSC can differentiate into myofibroblasts.

Our response: We thank for this comment. This is not our field of expertise, however as iPSCs can generate all other cell types than gametes, we believe it is possible. Mesenchymal stromal cells are derived from mesodermal lineage, meaning that the derivation is done at first to mesodermal progenitor cell via nodal, activin A and Wnt induction. Then the iPSC-mesodermal progenitor cells are directed towards cardiac mesodermal progenitor via FGF and BMB2 induction and notch inhibition. If wishing to generate fibroblasts, then the route would be via BMP and Wnt induction to epicardial progenitor and thereafter TGFβ induction towards fibroblast. However, as the myofibroblasts are specific type of muscle cells then the differentiation route might be after epicardial progenitor cell with TGFβ and FGF induction towards smooth muscle cell. But if taking the differentiation pathway from mesodermal progenitor cell towards multipotent cardiac progenitor cell and thereafter to muscle cell, then the induction would be done with TGFβ and PDGF-BB. At least according to the protocol of Doyle at al 2015 (DOI 10.1007/s12015-015-9596-6), however after this there has been several refinements of the protocol.

Minor point

  • abstract line 12: two-dimensional 2D -> two-dimensional (2D)

Our response: Thank you for the acuity. Brackets now added around 2D. In the final form of the manuscript on row 12.

  • page 8, 232: Ct:s, Please make it complete.

Our response: Yes, it is true that we did not explain what Ct stands for. Now corrected to form “cycle threshold (Ct) value”. In the final form of the manuscript on row 245.
